# SARS-CoV-2 Antibody Screening in Healthcare Workers in Non-Infectious Hospitals in Two Different Regions of Southern Poland (Upper Silesia and Opole Voivodeships): A Prospective Cohort Study

**DOI:** 10.3390/ijerph18084376

**Published:** 2021-04-20

**Authors:** Rafał Jakub Bułdak, Elżbieta Woźniak-Grygiel, Marta Wąsik, Janusz Kasperczyk, Ewa Gawrylak-Dryja, Renata Mond-Paszek, Adam Konka, Karina Badura-Brzoza, Martyna Fronczek, Marlena Golec, Mateusz Lejawa, Marcin Markiel, Sławomir Kasperczyk, Zenon Brzoza

**Affiliations:** 1Department of Clinical Biochemistry and Laboratory Diagnostics, Institute of Medical Sciences, University of Opole, Oleska 48, 45-052 Opole, Poland; marta.wasik@uni.opole.pl (M.W.); ewa.gawrylak-dryja@uni.opole.pl (E.G.-D.); renata.mond-paszek@uni.opole.pl (R.M.-P.); 2Silesian Park of Medical Technology Kardio-Med Silesia, M. Curie-Skłodowskiej 10C, 41-800 Zabrze, Poland; a.konka@kmptm.pl (A.K.); m.fronczek@kmptm.pl (M.F.); m.golec@kmptm.pl (M.G.); m.lejawa@kmptm.pl (M.L.); 3Department of Histology, Institute of Medical Sciences, University of Opole, Oleska 48, 45-052 Opole, Poland; ewozniakg@uni.opole.pl; 4Department of Environmental Medicine and Epidemiology, School of Medicine with the Division of Dentistry in Zabrze, Medical University of Silesia in Katowice, Jordana 18, 41-808 Zabrze, Poland; jkasperczyk@sum.edu.pl; 5Department of Psychiatry in Tarnowskie Góry, School of Medicine with the Division of Dentistry in Zabrze, Medical University of Silesia in Katowice, 42-612 Tarnowskie Góry, Poland; kbbrzoza@sum.edu.pl; 6Department of Pharmacology, Faculty of Medical Sciences in Zabrze, Medical University of Silesia in Katowice, Jordana 38, 41-808 Zabrze, Poland; 7Intensive Care Unit, Regional Specialised Hospital No. 4 in Bytom, al. Legionów 10, 41-902 Bytom, Poland; marcin.markie@gmail.com; 8Department of Biochemistry, Faculty of Medical Sciences in Zabrze, Medical University of Silesia in Katowice, 40-055 Katowice, Poland; skasperczyk@sum.edu.pl; 9Department of Internal Diseases, Institute of Medical Sciences, University of Opole, Al. W. Witosa 26, 40-451 Opole, Poland; zenon.brzoza@uni.opole.pl

**Keywords:** SARS-CoV-2, COVID-19, RT-PCR, asymptomatic, health-care professionals, serological surveillance, antibody screening, immunoglobulins

## Abstract

(1) Background: Detection of asymptomatic or subclinical human coronavirus SARS-CoV-2 infection in healthcare workers (HCWs) is crucial for understanding the overall prevalence of the new coronavirus and its infection potential in public (non-infectious) healthcare units with emergency wards. (2) Methods: We evaluated the host serologic responses, measured with semi-quantitative ELISA tests (IgA, IgG, IgM abs) in sera of 90 individuals in Hospital no. 4 in Bytom, 84 HCWs in the University Hospital in Opole and 25 in a Miasteczko Śląskie local surgery. All volunteers had negative RT-PCR test results or had not had the RT-PCR test performed within 30 days before sampling. The ELISA test was made at two different time points (July/August 2020) with a 2-weeks gap between blood collections to avoid the “serological window” period. (3) Results: The IgG seropositivity of asymptomatic HCWs varied between 1.2% to 10% (Opole vs. Bytom, *p* < 0.05; all without any symptoms). IgA seropositivity in HCWs was 8.8% in Opole and 7.14% in Bytom. IgM positive levels in HCWs in Opole and Bytom was 1.11% vs. 2.38%, respectively. Individuals with IgA and IgM seropositivity results were observed only in Opole (1.19%). More studies are needed to determine whether these results are generalizable to other populations and geographic as well as socio-demographic locations. (4) Conclusions: 100% of IgG(+) volunteers were free from any symptoms of infection in the 30 days before first or second blood collection and they had no awareness of SARS-CoV-2 infection. Asymptomatic HCWs could spread SARS-CoV-2 infection to other employees and patients. Only regular HCWs RT-PCR testing can reduce the risk of SARS-CoV-2 spreading in a hospital environment. The benefit of combining the detection of specific IgA with that of combined specific IgM/IgG is still uncertain.

## 1. Introduction

The rapid spread of SARS-CoV-2 coronavirus infections caused the World Health Organization to declare the COVID-19 pandemic on 11 March 2020. Since mid-December 2019 to early April 2021, over 130.5 million cases of COVID-19 and 2.8 million deaths caused by coronavirus SARS-CoV-2 infection have been reported globally [1]. At the beginning of September 2020, the daily number of confirmed cases of COVID-19 in Poland began to increase significantly [2]. The current situation has entered a phase that is very difficult to control. Dominant points of infections are numerous and small, rather than isolated and big as was the case at the beginning of the pandemic.

Consistent criteria for case definition, COVID-19 diagnosis in suspected cases, and scaling up of suitable diagnostic systems have been challenging since the start of the pandemic. Reverse transcription polymerase chain reaction (RT-PCR)-based nasopharyngeal swab testing was rapidly developed and has helped in confirming and tracking the SARS-CoV-2 RNA [3]. The sensitivity of the RT-PCR method depends on the timing and time of respiratory sampling. Serological testing for SARS-CoV-2-specific immunoglobulins (Ig) is relatively easy, inexpensive and critical for epidemiological studies. SARS-CoV-2 specific B cell responses appear to correlate with disease severity, with rising antibody titres between 5 to 10 days and fully positive rates at about 18 days after symptom onset [3]. The present serological research was performed during the first wave of the rise of morbidity in Poland, where most institutions focused on genetic diagnosis of SARS-CoV-2 infection (vide RT-PCR) to eliminate infected health care workers (HCWs) from the health care system (HCS).

HCWs are the frontline workforce for the clinical care of suspected and confirmed SARS-CoV-2 cases. They are presumably exposed to a higher risk of contracting the disease than the general population. Protection of HCWs and their families from contracting COVID-19 in hospitals is paramount and underscored by rising numbers of HCW deaths nationally and internationally [4].

During the rapid spread of the COVID-19 pandemic, 2020 was marked by ongoing research into the disease. Research has also focused on SARS-CoV-2 antibodies detection in the blood of HCWs [5,6,7,8,9,10,11]. When estimating disease transmission in a specific occupational setting, only serological tests performed on two blood samples from the same volunteers 2 weeks apart can be fully reliable. Our immunoassays were performed in so-called twin (even) sera to eliminate the risk of serological window, which significantly increases the diagnostic reliability of the present results. The majority of the published papers include serological data obtained from single serum sample collection [4,12,13,14,15].

Here, we aimed to determine the serological status of HCWs who were not aware of prior SARS-CoV-2 infection or any other disease symptoms in selected non-infectious hospitals during the first wave of the epidemic in the southern region of Poland.

Detailed aims of the study:-Did the percentage of IgG (+) workers differ across hospitals in two different voivodeships, the voivodeships showing a tenfold difference in the number of new (daily) infections?-Did IgM and/or IgA (+) volunteers have active SARS-CoV-2 viremia?-Did seropositive volunteers included in the study exhibit symptoms of respiratory tract infection 30 days prior to serological tests?

## 2. Materials and Methods

This study was reviewed and approved by the Medical Ethical Committee of the Medical University of Silesia by resolution No. PCN/0022/KB1/50/20.

### 2.1. Data Extraction and Analysis

Swab result data was extracted from the hospital-laboratory interface software. Details of symptoms were extracted manually from a diagnostic survey (questionnaires). Data was collated using Microsoft Excel. Statistical calculations were analyzed in the STATISTICA program. The Chi-squared test was used for the comparison of positive rates between HCWs in the surveyed institutions defined in the main text. A significant *p*-value was assumed at the level of *p* < 0.05. Mann-Whitney testing was used to compare absorbance values between different categories of tested individuals.

### 2.2. Patients

We invited HCWs in three medical facilities in Poland (University Hospital in Opole; Hospital no. 4 in Bytom and Eko-Prof-Med Healthcare Unit (HCU) in Miasteczko Śląskie) to determine seroprevalence of anti-SARS-CoV-2 antibodies.

Out of 1650 HCWs of three healthcare units, 225 (13.6%) agreed to participate in the study, out of which 199 (12.1%) were recruited and were included in the study (Figure 1). The applicants were assessed with inclusion and exclusion criteria. We adopted very restrictive criteria for inclusion in the study, the main criteria being: continuity of work at least 6 weeks (in the first wave, with the paralysis of health care, there were huge staff shortages that eliminated further potential participants for inclusion in the study). The remaining inclusion criteria for the study were for volunteers to have signed a form of informed consent to participate in a medical experiment project and to have enjoyed good or very good health status. The study excluded volunteers with a positive SARS-CoV-2 test by RT-PCR, with an active form of respiratory tract infection, ones with a CRP (C-reactive protein) result above the reference value, ones of generally poor health (subjective assessment of the subject), ones who had not agreed to participate in the study, those who had not completed the diagnostic questionnaire, ones who lacked continuity in professional activity within the period of 6 weeks prior to recruitment to the medical experiment, and volunteers under 18 years of age. We did not include volunteers who were RT-PCR positive in the analysis because 100% of (+) RT-PCR patients are IgG seropositive 10–19 days after infection [16,17]. Thus, we would have overestimated the numbers and we only wanted to screen active HCWs who did not know whether they had contracted SARS-CoV-2 infection (the social aspect of our study), especially since in addition to the scientific aspect of the result, the volunteers received a diagnostically reliable test result (double blood draw to avoid the risk of a serological window).

Each individual in the HCW screening group was contacted by telephone, e-mail or project web page (https://badania.labcovid-19.pl/pl, accessed on 1 July 2020) to establish their clinical history and COVID-19 probability criteria. The clinical criteria for estimating pre-test probability of COVID-19 in HCWs were fever (>38 Celsius degrees), cough, sore throat, shortness of breath, nausea and/or vomiting, loss of smell and/or taste. A diagnostic questionnaire (with questions about their health status) was filled in by the volunteers using a web page or a web application: https://badania.labcovid-19.pl/pl/eksperyment-medyczny-formularz-rekrutacyjny (accessed on 1 July 2020).

Positive results (IgA, IgM) from the first ($n_{I}$) or the second ($n_{II}$) blood collection were telephoned to volunteers by hospital physicians, who took further details about the interpretation of the results. IgA and IgM positive volunteers were examined for the presence of SARS-CoV-2 RNA coronavirus with the RT-PCR method (see Figure 1). After completing the present study had been completed, all participants were informed about test results with data interpretation.

Reports from the current study will be presented to local sanitary-epidemiological stations in Katowice and Opole as well as to local authorities after publication of the present data.

### 2.3. Antibody Measurement

The enzyme linked immunosorbent assay (ELISA) test was made at two different time points (July/August 2020) with a 2-weeks gap between blood collections to avoid the “serological window” period. The IgG, IgM and IgA antibodies against SARS-CoV-2 in serum samples were tested using enzyme linked immunosorbent assay (ELISA) kits supplied by Euroimmun Medizinische Labordiagnostik [Anti-SARS-CoV-2 ELISA (IgG); Anti-SARS-CoV-2 NCP ELISA (IgM) and Anti-SARS-CoV-2 ELISA (IgA)], in line with the manufacturer’s instructions. To put it briefly, the test kits contain microplate strips with reagent wells coated with modified nucleo-capsid protein of SARS-CoV-2 to detect IgM antibodies, and with recombinant structural protein (S1 domain) of SARS-CoV-2 to detect IgG and IgA antibodies. Diluted patient’s samples are incubated in the wells. In the case of positive samples, specific IgG, IgA or IgG antibodies bind the antigens. To detect the bound antibodies, a second incubation is carried out using an enzyme-labelled anti-human IgM catalysing a colour reaction with chromogen solution. In samples taken prior to day 10 (time point after onset of symptoms or positive RT-PCR), the Anti-SARS-CoV-2 NCP ELISA (IgM) was shown by the manufacturer to provide a sensitivity of 88.2%. The sensitivity of the Anti-SARS-CoV-2-NCP-ELISA (IgM) for samples collected in the period from day 11 to 15 is 70.6%. The determined positive results correspond to a 99% specificity of the Anti-SARS-CoV-2 ELISA (IgG). The determined positive results correspond to an 88.4% specificity of the Anti-SARS-CoV-2 ELISA (IgA).

## 3. Results

A total of 199 healthy participants were included in the study (Figure 1: 84 (42.2%) from the University Hospital in Opole, 90 (45.2% of the overall number of HCWs) from Hospital No. 4 in Bytom and 25 (12.6%) from the Eko-Prof-Med HCU in Miasteczko Śląskie. Demographic characteristics of participants included in the study are shown in Table 1. The group included nurses (44.7%), physicians (29.7%), laboratory diagnosticians (9.1%), paramedics (4.5%) and other medical staff (12.0%). On the day of recruitment, the volunteers showed no symptoms of infection. The median age (IQR) was 44.5 (37–50) for the individuals from the University Hospital in Opole, 47.5 (38–53) for the individuals from Hospital No. 4 in Bytom and 51.5 (46.5–56) for the individuals from the Eko-Prof-Med HCU in Miasteczko Śląskie. In each of the studied groups, men constituted a definite minority. Among the recruited participants, 27 (32.1%) men were recruited from the Clinical University Hospital in Opole, 10 (11.1%) men were recruited from Hospital No. 4 in Bytom and 3 (12.0%) men were recruited from the Eko-Prof-Med HCU in Miasteczko Śląskie. Demographic characteristics of the participants in the study are presented in Table 1. In the University Hospital in Opole the number of positive results in the IgA/IgG/IgM antibody class was 5.95%/1.19%/2.38% and borderline ratio for IgA/IgG/IgM was 1.19%/0.00%/1.19% (Table 2). In the case of Hospital No. 4 in Bytom the seropositivity in the IgA/IgG/IgM antibody class was 8.89%/10.00%/1.11% with borderline ratio 1.11%/0.00%/0.00% (Table 3). In the analysis of samples from the Eko-Prof-Med HCU the positive result in the IgA/IgG/IgM antibody class was 4.00%/4.00%/0.00% and the borderline ratio for IgA/IgG/IgM was 0.00%/0.00%/0.00% (Table 4).

IgA and IgM positive HCWs were examined for the presence of the SARS-CoV-2 RNA coronavirus using the RT-PCR method (in Figure 1). All the IgA and IgM positive volunteers were RT-PCR SARS-CoV-2 negative. Moreover, statistical analysis showed that positive antibody scores across all classes had no statistical significance in relation to the symptoms that were taken into account in the participant questionnaire (Table 5).

The results of statistical analysis are presented in Table 5. The differences between the percentage distribution of anti-SARS-CoV-2 IgG positive results across institutions (Bytom 10.0%, Miasteczko Śląskie 4.0%, Opole 1.2%) have shown a statistically significant difference between the Opole and Bytom hospitals (*p* = 0.0127). Comparing the antibodies test results in the same classes in the first ($n_{I}$) and the second ($n_{II}$) blood collection, statistically significant differences were found between the positive results in all classes (IgA $n_{I}$/IgA $n_{II}$; IgG $n_{I}$/IgG $n_{II}$; IgM $n_{I}$/IgM $n_{II}$). Statistically significant differences were also found between IgA and IgG classes, both in the first and second blood collection (IgA $n_{I}$/IgG $n_{I}$; IgA $n_{II}$/IgG $n_{II}$).

## 4. Discussion

Serological testing is widely proposed as a major tool to manage the coronavirus disease 2019 (COVID-19) pandemic, playing a key role in more accurate disease burden assessment, identification of potential donors for therapeutic immune plasma, and tracking evolution toward population-level immunity [18]. All patients with symptomatic infection develop detectable immunoglobulin G (IgG) antibodies within 14–19 days of symptom onset, consistent with patterns seen in other systemic viral infections [16,17]. Other authors also demonstrated that IgG seroconversion was commonly observed in both symptomatic and asymptomatic COVID-19 patients [19,20,21].

This SARS-CoV-2 antibodies screening study included HCWs recruited in two non-infectious COVID-19 units in two different regions of southern Poland. The analysis included volunteers who had negative RT-PCR test results or had not had the RT-PCR test performed within 30 days before the beginning of the study. Immunoglobulin G (IgG antibodies) have an important role in the neutralization of SARS-CoV-2 [22]. Therefore, an IgG response indicates COVID-19 disease, also asymptomatic. The low rate of IgG seropositive results in HCWs in the University Hospital in Opole may be partly explained by lower rates of infection in the Opole region in comparison with the Upper Silesia region; (cumulative incidence 0.05% vs. 0.5%; respectively, thus far); (Figure 2 and Figure 3). It can be explained by the fact that the number of RT-PCR tests performed in the Opole region between July to August was also significantly lower, by about 13, than in the Upper Silesia region (Table 6). Another difference between those two regions is the socio-demographic aspect. The Upper Silesia region has a higher concentration of the population than the Opole region. It should be noted that population density in Poland is 123 persons per square kilometer. According to the statistical information data published in 2020 by Statistics Poland, population density in the Silesian voivodeship is 366 people per square kilometer, while in the Opolskie voivodeship this value is below 105 people per square kilometer [23]. Moreover, the urbanization rate of the Silesian voivodeship is higher when compared to the Opolskie voivodeship. The Upper Silesia region is also characterized by a higher level of industrialization (high values of industrial infrastructure density and a higher share of employment in industry) compared to the Opole voivodeship [24]. These differences in population density, urbanization and industrialization rates between the studied regions may explain the difference in the number of infections.

The differences in the percentage of IgG(+) HCWs in Bytom hospital vs. Opole may also be related to the different health strategies presented by these two medical centers. HCWs from Opole were tested regularly every four weeks, while HCWs from Bytom were tested only intermittently after outbreaks of symptomatic unit infections in the hospital. The regularity of genetic testing among HCWs and the fear of a positive RT-PCR test may result in greater self-discipline among health personnel in adhering to sanitation regimens and the use of personal protective equipment. Based on analysis of the responses of the volunteers included in the present study, no IgG(+) individuals indicated any symptoms of infection.

Our data shows that focusing solely on the testing of individuals fitting a strict clinical case definition for COVID-19 will inevitably miss asymptomatic infected individuals. In our study, on the day of recruitment the individuals showed no symptoms of infection (negative answers in the questionnaire: low-grade or high-grade fever, cough, changes in smell and taste, digestive system disorders, respiratory infection symptoms). Despite this, 11 individuals from the study population showed a positive result of the IgG antibody class in both blood samplings. This strongly supports the existence of asymptomatic spreaders: people who are asymptomatic but infected, and infect other workers (and others they have contacted, including their families). In the health questionnaire, a single asymptomatic person indicated that they had had digestive system disorders in the last 30 days. However, after analyzing all the responses in this case, we concluded that it was an isolated symptom that did not indicate a systemic infection. In a study from India researchers found that just 5% of people accounted for 80 percent of infections detected by contact tracing [25]. Furthermore, 71% of people did not transmit the virus to other people. Therefore, a very small percentage of the infected are responsible for transmitting the virus to other people. This is why it is so important to regularly test people, also the asymptomatic, as this helps to isolate asymptomatic infected people from the healthy population [29]. Chen et al., 2020 also revealed that the serological testing is useful for the identification of asymptomatic or subclinical infection of SARS-CoV-2 among close contacts with COVID-19 patients. The author’s analysis of the hospital setting highlighted a high percentage of IgG (17.4%, *n* = 105) of asymptomatic or subclinical SARS-CoV-2 infection in HCWs during the first wave of SARS-CoV-2 outbreak in China [12].

The benefit of combining the detection of specific IgA with that of combined specific IgM/IgG is still uncertain. According to the literature, detectable IgM usually precedes IgG, some patients show simultaneous rises in both antibodies, and the intensity of responses is heterogeneous [18]. On the other hand, all IgA and IgM positive HCWs presented in our study were examined for the presence of the SARS-CoV-2 RNA coronavirus using the RT-PCR method. Their swab samples (100%) were found to be SARS-CoV-2 RNA negative, thereby diagnostic utility of these abs in detection of acute SARS-CoV-2 infection was not confirmed. Secondly, in our study, all positive IgA or IgM sera were obtained from asymptomatic HCWs. Mucosal and systemic IgA responses that may play critical roles in the COVID-19 pathogenesis have received much less attention. Others authors observed correlation between IgA serum level and disease severity of RT-PCR (+) confirmed patients. Yu et al., 2020 confirmed significant differences in the relative levels of IgA and IgG between severe and non-severe patients after the disease onset. They also found a significant positive association of SARS-CoV-2 specific IgA level and the APACHE-II score in critically ill patients with COVID-19 [30]. On the contrary, Huang et al., 2020 demonstrated that anti-SARS-CoV-2 serum IgA may appear before anti-SARS-CoV-2 IgG and that IgA titer appears higher in patients with severe COVID-19 compared to those with milder illness [31]. All the above mentioned associations between high serum IgA levels were described in patients with positive RT-PCR test result [17,31]. On the contrary, in our study, all HCWs volunteers enrolled in analysis had negative RT-PCR test results or not had the RT-PCR test performed within 30 days before sampling. We did not include RT-PCR (+) patients in the analysis because 100% of these patients show the presence of IgG antibodies 14–19 days after infection [17,18]. Moreover, statistical analysis showed that positive antibody scores across all classes had no statistical significance in relation to the symptoms that were taken into account in the participant questionnaire (Table 5). Secondly, our immunoenzymatic reagent wells used in the present study were coated with (strong immunogenic) recombinant structural protein (S1 domain) of SARS-CoV-2 to detect IgA and IgG abs unlike research conducted by Huang et al. and Guo et al. (2020), which performed tests using kits containing modified nucleo-capsid protein (N) of SARS-CoV-2 (less immunogenic) to detect IgA and IgM abs. These authors concluded that IgA as well as IgM detection can aid in diagnosis of COVID-19 including subclinical cases [13,28]. Therefore, results obtained on the basis of serological tests with different epitopes (i.e., S vs. N), which show different immunogenicity, should not be compared.

Unlike RT-PCR tests which are highly specific, cross-reactivity is a major challenge for COVID-19 antibody tests. Let us consider the occurrence of cross-reactivity that may affect the results of the SARS-CoV-2 antibodies tests, especially IgA or/and IgM abs in our study. In the case of IgM antibodies, the manufacturer explains that thanks to the use of a modified nucleocapsid protein, in which significant homologous regions were eliminated and the diagnostically relevant epitopes were combined, cross-reactions with most human pathogenic representatives of the coronavirus family are virtually excluded. Cross-reactions between SARS-CoV(-1) and SARS-CoV-2, however, are likely to occur due to their close relationship. The immunoglobulin class IgG and IgA against SARS-CoV-2 to most of the human pathogenic representatives of this virus family were virtually excluded because of low homologies of the S1 protein within the coronavirus family. The close relationship of SARS-CoV(-1) and SARS-CoV-2 may cause crossreactions between these two viruses. Sera from patients with SARS-CoV(-1), MERS-CoV, HCoV-229E, HCoV-NL63, HCoV-HKU1, or HCoV-OC43 infections were investigated to examine this further. Pronounced cross-reactions occur mainly with Anti-SARS-CoV(-1) IgG antibodies. Cross-reactions to other human pathogenic coronaviruses were not observed by the manufacturer.

According to the manufacturer (instruction for use), cross-reactivity causes 3.4% and 8.6% of people vaccinated against influenza to test false-positive for SARS-CoV-2 antibodies in the blood (anti-SARS-CoV-2 IgG and anti-SARS-CoV-2 IgA, respectively). Samples that were positive for antibodies against influenza, including the freshly vaccinated, did not affect the specificity of the test of the anti-SARS-CoV-2 IgM (in this case the manufacturer gives 100% specificity). In our study, the percentage of the cohort vaccinated against influenza (in the last 12 months) was 28 individuals (14.07% of the study population). The number of people showing a positive result in the IgA/IgG/IgM antibody class in the first ($n_{I}$) and/or second ($n_{II}$) blood sampling and simultaneously vaccinated against influenza was: 7 people (1 person in the IgA and 1 person in the IgM antibody class in Opole; 2 people in IgA, 1 person in IgG in Bytom; 1 person in the IgA and 1 person in the IgG in Miasteczko Śląskie); (data not shown). In our opinion, the number of people who tested positive for the surveyed classes of antibodies and simultaneously vaccinated against influenza is too small, to conclude whether there is a correlation in cross-reactivity. The aspect of cross-reactivity in individuals vaccinated against influenza and testing positive for coronavirus should be taken into account in studies on very large groups. However, in the first months of 2021, we will repeat the study of coronavirus antibodies in the same research group. With a more extensively studied population, we should show a greater number of people with a positive result in any class of antibodies. With a larger group of people studied, it may then be possible to draw statistically significant conclusions about the occurrence of cross-reactivity after vaccination against influenza.

Brett and Rohani note that achieving herd immunity is unrealizable as the health policies to be pursued would need to be very restrictive [32]. This would require a strict implementation of constraints such as social distancing and lockdowns over a longer period to avoid overloading the healthcare system. However, the authors point out that an effective inhibition of the COVID-19 pandemic is quite possible within two months with strict social distancing and isolation of infected people. The authors point out that periodic screening tests are necessary to identify infected people. Risk analysis performed by Chen et al., 2020 revealed that wearing a face mask could reduce the infection risk in hospital environments [12].

We observed the differences between the percentage distribution of IgG positive results in the University Hospital in Opole and Hospital no. 4 in Bytom (1.2% vs. 10% respectively). Employees of the hospital in Opole had been regularly tested with the RT-PCR test (every 4 weeks), unlike the employees of the other facilities surveyed, which could have had an impact on the identification of asymptomatic carriers of COVID-19. The psychological aspect (to avoiding infection) is also very important—awareness that individuals will be tested may increase observance of the sanitary regime in comparison with individuals that will not be tested.

We suppose that the transmission of SARS-CoV-2 infections at Hospital No. 4 in Bytom could have been the result of the exposure of staff and/or patients to so-called super-spreaders. We are not able to show directly that super-spreaders are responsible for outbreaks of disease in hospitals, but the difference in the numbers of Ig (+) positive individuals found in the study in both centers (Bytom vs. Opole) indirectly indicates such a possibility. The SARS-CoV-2 coronavirus RNA diagnosis strategy for healthcare professionals should include genetic testing of all hospital healthcare professionals (especially in emergency wards) regardless of whether they show symptoms. In the authors’ opinion, such a strategy (testing regardless of presence/lack of symptoms) can protect medical personnel and the functioning of hospitals more effectively, as asymptomatic members of staff who remain professionally active can transmit the infection to other medical personnel and patients. Similarly, in a prospective cohort of 829 individuals without previous diagnoses of SARS-CoV-2 infection or COVID-19, 7.3% of HCWs and 0.4% of non-HCWs (NHCWs) were found to be IgG positive in two large U.S. universities and two affiliated university hospitals. These results support the hypothesis of higher SARS-CoV-2 prevalence in HCWs compared with NHCWs, a difference which may be attributable to workplace exposures, given the low rate of infection in NHCWs [33]. Garcia-Basteiro et al., 2020 found that 9.3% of HCW from a large Spanish referral hospital during the first wave of COVID-19 outbreak developed detectable IgA, IgG, and/or IgM antibodies. Combining data from antibody detection and previous or current positive RT-PCR, the cumulative prevalence of SARS-CoV-2 infection rose to 11.2%. However, 40.0% of the seropositive HCW had not previously been diagnosed with COVID-19 and 23.1% were asymptomatic, indicating that a large percentage of infections were undetected [4]. The majority of research with SARS-CoV-2 antibody detection was conducted solely from a single blood sampling collected from HCWs, which including both RT-PCR positive and negative volunteers [4,12,13,14,15]. Only a few SARS-CoV-2 serological studies including ours were performed on patient’s sera collected at least two times with gap window period between blood collection [18,19].

According to the authors, with the changing strategy of genetic testing of the patient and an increased participation of antigen tests in the diagnosis of SARS-CoV-2 with lower sensitivity and specificity of these tests (especially in the group of asymptomatic patients or the group of patients with symptoms, when the test was performed in the first two days after the onset of symptoms), there is a serious risk of overlooking false-negative individuals who are infected and have the potential to transmit infection. While this approach to testing the general population for non-ambulatory treatment may only falsify the actual number of fresh infections, in the case of the using this strategy for the testing of key personnel, it may not be sufficient, because testing of key personnel should be based on the methods of diagnosis which have the highest sensitivity and specificity (RT-PCR). Only in this way can (constant and symptom-independent) testing of key personnel eliminate the risk caused by the presence of infected workers RT-PCR (+), who may transmit infection in the work environment. Our study included significantly more volunteers than qualified for the analysis. New infections, holiday periods or layoffs and absenteeism from work, two blood draws (with a 14-day break in between) resulted in a reduction in the final number of volunteers included in the analysis for reasons beyond the control of the researcher. Another limitation was the fact that the vaccination preventive programme for HCWs (group “0”) was introduced from 27 December 2020, which made it impossible to perform subsequent blood draws on the same volunteers. The exception was the health center in Miasteczko Śląskie (without emergency ward), where every second HCWs was randomly examined. Xu et al., 2020 revealed that IgG seropositivity of HCWs differed among staff working with direct patient contacts (34.7%) when compared to medical personnel working in a non-clinical hospital without an emergency ward (22.6%) [34].

Future perspectives: In the first months of 2021, we will repeat the study of coronavirus antibodies in the same research group before vaccination. Following the vaccination programme for the HCW group, we will quantify IgG class antibody levels in the same volunteers who are willing to be vaccinated, and in new recruits.

## 5. Conclusions

-HCWs with a positive IgG (+) test result were unaware that they were infected with SARS-CoV-2 and may have unwittingly contributed to transmission of the virus in the hospital environment.-Only regular screening RT-PCR tests among healthcare personnel, regardless of whether they show signs of infection, can be an effective way to prevent virus transmission to others in their immediate environment.-The benefit of combining the detection of specific IgA with that of combined specific IgM/IgG is still uncertain.

In summary, our data suggests that the roll-out of screening programmes to include asymptomatic as well as symptomatic patient-facing staff should be a national priority to limit avoidable SARS-CoV-2 transmission in hospital environment. Such an approach will be critical for protecting patients and hospital staff from the next waves of SARS-CoV-2 pandemic.

## Figures and Tables

**Figure 1 ijerph-18-04376-f001:**
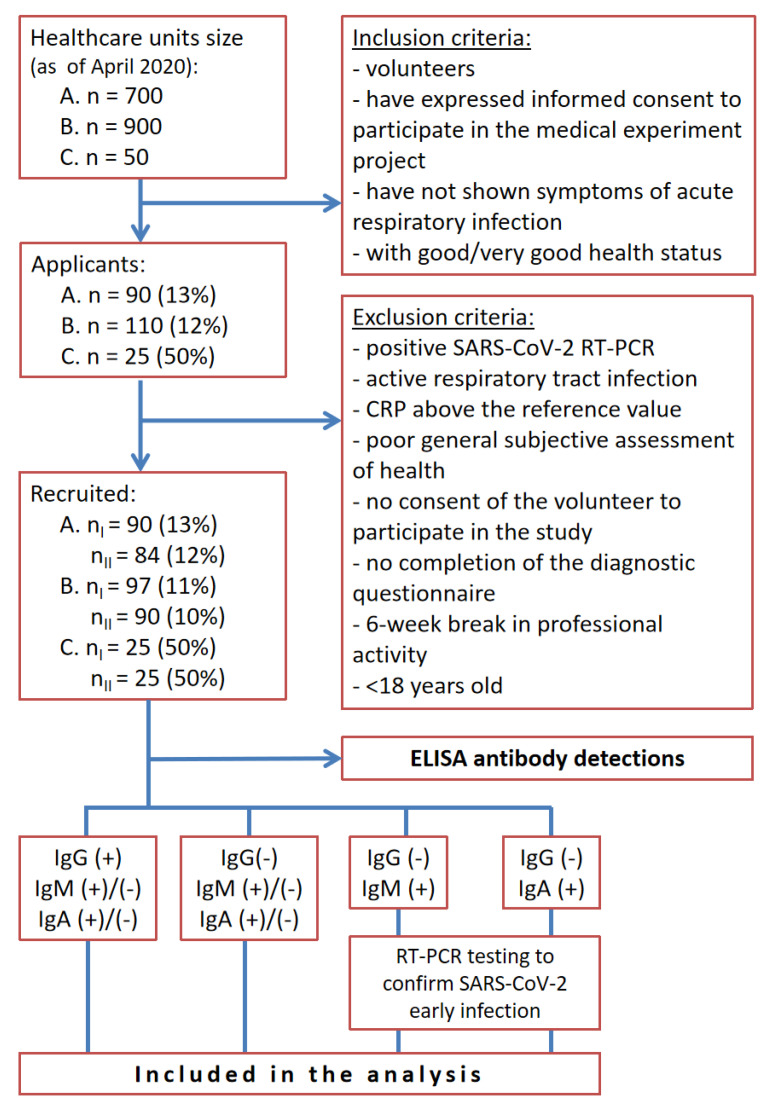
Study design flowchart in three Polish medical institutions (A.-University Hospital in Opole; B.-Hospital no. 4 in Bytom and C.-Eko-Prof-Med Healthcare Unit (HCU) in Miasteczko Śląskie; HCWs-health care workers; $n_{I}$-first blood collection; $n_{II}$-second blood collection; RT-PCR-Reverse Transcription Polymerase Chain Reaction), CRP-C-reactive protein.

**Figure 2 ijerph-18-04376-f002:**
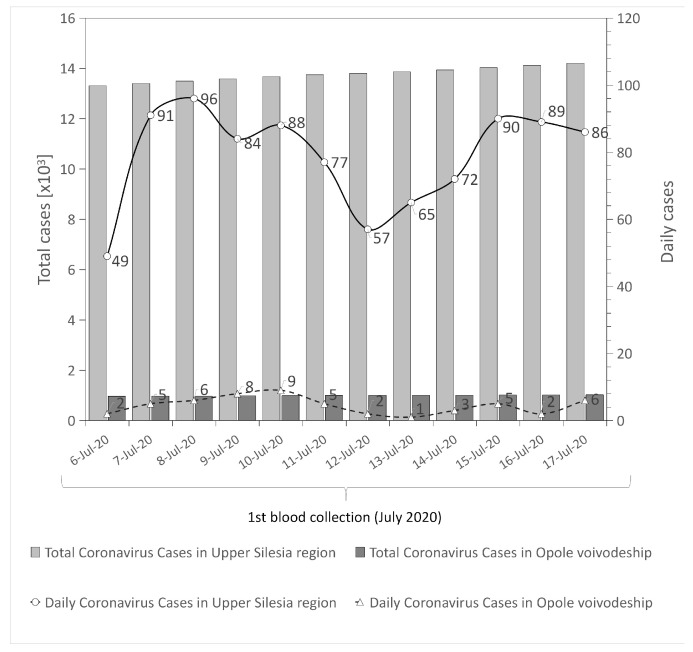
Number of infected individuals in both regions during 1st blood collection ($n_{I}$, July 2020) [25,26,27,28].

**Figure 3 ijerph-18-04376-f003:**
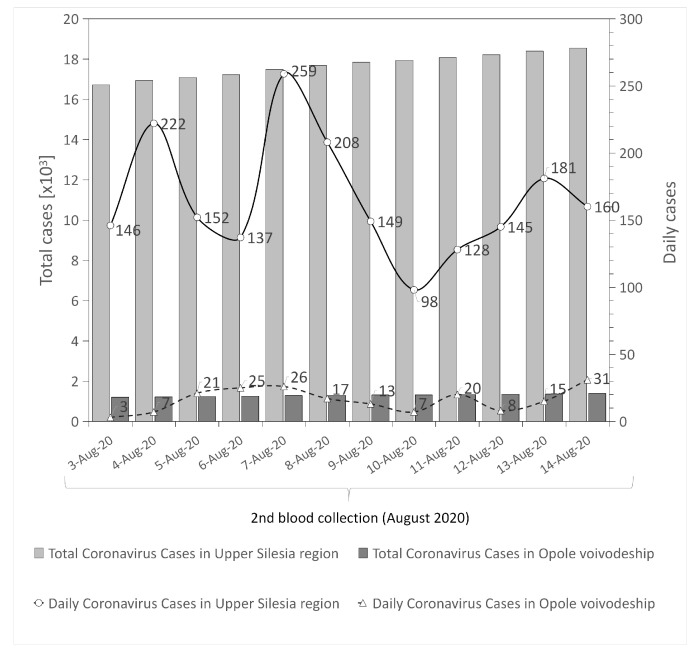
Number of infected individuals in both regions during 2nd blood collection ($n_{II}$, August 2020) [25,26,27,28].

**Table 1 ijerph-18-04376-t001:** Demographic characteristics of Healthcare Units (HCUs) included in the study.

				Time of Blood Collection				
**Population**	**Name of HCU/City**	**Emergency Ward**	**Geographic Region of Poland**	**$n_{I}$**	**$n_{\mathrm{II}}$**	**Number of HCWs Included in Analysis (% of Overall Employed)**	**Median Age, Year $Q_{1}$ = 25th Percentile, $Q_{2}$ = Median, 50th Percentile, $Q_{3}$ = 75th Percentile**	**Male %**
HCWs	Clinical University Hospital in Opole	Yes	Opole voivodeship	90	84	84 (12)	$Q_{1}$ = 37; $Q_{2}$ = 44.5; $Q_{3}$ = 50	32.1
	Hospital no. 4 in Bytom	Yes	Silesian voivodeship	97	90	90 (10)	$Q_{1}$ = 38; $Q_{2}$ = 47.5; $Q_{3}$ = 53	11.11
	Eko-Prof-Med HCU in Miasteczko Śląskie	No		25	25	25(50)	$Q_{1}$ = 46.5; $Q_{2}$ = 51.5; $Q_{3}$ = 56	12

**Table 2 ijerph-18-04376-t002:** Positive rates of the serological tests in HCWs* included in the study from the University Hospital in Opole.

ELISA Test (Serological Assay) Outcomes OPOLE, Opole Voivod.									
($n_{I}=90$)									
**IgA**		**IgG**		**IgM**		**IgA/IgG**		**IgA/IgM**	
(+)	BR	(+)	BR	(+)	BR	(+)	BR	(+)	BR
8	3	1	1	2	2	0	0	1	0
(8.89%)	(3.33%)	(1.11%)	(1.11%)	(2.22%)	(2.22%)	(0.00%)	(0.00%)	(1.11%)	(0.00%)
($n_{II}=84$)									
**IgA**		**IgG**		**IgM**		**IgA/IgG**		**IgA/IgM**	
(+)	BR	(+)	BR	(+)	BR	(+)	BR	(+)	BR
7	2	2	0	6	4	1	0	1	0
(8.33%)	(2.38%)	(2.38%)	(0.00%)	(7.14%)	(4.76%)	(1.19%)	(0.00%)	(1.19%)	(0.00%)
Analysis of two serial sera specimens (*n* = 84)									
**IgA**		**IgG**		**IgM**		**IgA/IgG**		**IgA/IgM**	
(+)	BR	(+)	BR	(+)	BR	(+)	BR	(+)	BR
5	1	1	0	2	1	0	0	1	0
(5.95%)	(1.19%)	(1.19%)	(0.00%)	(2.38%)	(1.19%)	(0.00%)	(0.00%)	(1.19%)	(0.00%)

* HCWs included nurses, physicians, laboratory diagnosticians, paramedics and other medical staff, (+)—seropositivity, BR—borderline ratio (+/−), $n_{I}$—first blood collection (July 2020), $n_{II}$—second blood collection (August 2020).

**Table 3 ijerph-18-04376-t003:** Positive rates of the serological assay in HCWs* included in the study from Hospital no. 4 in Bytom.

ELISA Test (Serological Assay) Outcomes BYTOM, Silesian Voivod.									
($n_{I}=97$)									
**IgA**		**IgG**		**IgM**		**IgA/IgG**		**IgA/IgM**	
(+)	BR	(+)	BR	(+)	BR	(+)	BR	(+)	BR
8	3	9	1	1	1	7	0	0	0
(8.28%)	(3.09%)	(9.28%)	(1.03%)	(1.03%)	(1.03%)	(7.22%)	(0.00%)	(0.00%)	(0.00%)
($n_{II}=90$)									
**IgA**		**IgG**		**IgM**		**IgA/IgG**		**IgA/IgM**	
(+)	BR	(+)	BR	(+)	BR	(+)	BR	(+)	BR
9	1	9	0	1	0	8	0	0	0
(10.00%)	(1.11%)	(10.00%)	(0.00%)	(1.11%)	(0.00%)	(8.89%)	(0.00%)	(0.00%)	(0.00%)
Collective analysis of both sera (*n* = 90)									
**IgA**		**IgG**		**IgM**		**IgA/IgG**		**IgA/IgM**	
(+)	BR	(+)	BR	(+)	BR	(+)	BR	(+)	BR
8	1	9	0	1	0	7	0	0	0
(8.89%)	(1.11%)	(10.00%)	(0.00%)	(1.11%)	(0.00%)	(7.78%)	(0.00%)	(0.00%)	(0.00%)

* HCWs included nurses, physicians, laboratory diagnosticians, paramedics and other medical staff, (+)—seropositivity, BR—borderline ratio (+/−), $n_{I}$—first blood collection (July 2020), $n_{II}$—second blood collection (August 2020).

**Table 4 ijerph-18-04376-t004:** Positive rates of the serological assay in HCWs* included in the study from Eko-Prof-Med HCU in Miasteczko Śląskie.

ELISA Test (Serological Assay) Outcomes MIASTECZKO ŚLĄSKIE, Silesian Voivod.									
($n_{I}=25$)									
**IgA**		**IgG**		**IgM**		**IgA/IgG**		**IgA/IgM**	
(+)	BR	(+)	BR	(+)	BR	(+)	BR	(+)	BR
2	0	1	0	0	0	0	0	0	0
(8.00%)	(0.00%)	(4.00%)	(0.00%)	(0.00%)	(0.00%)	(0.00%)	(0.00%)	(0.00%)	(0.00%)
($n_{II}=25$)									
**IgA**		**IgG**		**IgM**		**IgA/IgG**		**IgA/IgM**	
(+)	BR	(+)	BR	(+)	BR	(+)	BR	(+)	BR
2	3	2	0	1	0	1	0	1	0
(8.00%)	(12.00%)	(8.00%)	(0.00%)	(4.00%)	(0.00%)	(4.00%)	(0.00%)	(4.00%)	(0.00%)
Analysis of two serial sera specimens (*n* = 25)									
**IgA**		**IgG**		**IgM**		**IgA/IgG**		**IgA/IgM**	
(+)	BR	(+)	BR	(+)	BR	(+)	BR	(+)	BR
1	0	1	0	0	0	0	0	0	0
(4.00%)	(0.00%)	(4.00%)	(0.00%)	(0.00%)	(0.00%)	(0.00%)	(0.00%)	(0.00%)	(0.00%)

* HCWs included nurses, physicians, laboratory diagnosticians, paramedics and other medical staff, (+)—seropositivity, BR—borderline ratio (+/−), $n_{I}$—first blood collection (July 2020), $n_{II}$—second blood collection (August 2020).

**Table 5 ijerph-18-04376-t005:** Observed number of individuals antibody-positive in different classes and analysis of the relationships between antibody classes and reported symptoms.

Classes of Antibodies	Classes of Antibodies					
	IgA $n_{I}$ (+)	IgA $n_{\mathrm{II}}$ (+)	IgG $n_{I}$ (+)	IgG $n_{\mathrm{II}}$ (+)	IgM $n_{I}$ (+)	IgM $n_{\mathrm{II}}$ (+)
IgA $n_{I}$ (+)		14; *p* = 0.000	7; *p* = 0.000		1; *p* = 0.120	
IgA $n_{II}$ (+)				*10; p = 0.000*		2; *p* = 0.108
IgG $n_{I}$ (+)				11; *p* = 0.000	0; *p* = 0.683	
IgG $n_{II}$ (+)						1; *p* = 0.486
IgM $n_{I}$ (+)						3; *p* = 0.000
IgM $n_{II}$ (+)						
**Classes of antibodies**	**Symptoms**					
	low-/high-grade fever (+)	cough (+)	changes in smell and taste (+)	digestive system disorders (+)	respiratory infection (+)	
IgA $n_{I}$ (+)	0; *p* = 0.602	0; *p* = 0.460	0; *p* = 0.672	1; *p* = 0.543	0; *p* = 0.548	
IgA $n_{II}$ (+)						
IgG $n_{I}$ (+)	0; *p* = 0.660	0; *p* = 0.545	0; *p* = 0.730	1; *p* = 0.300	0; *p* = 0.623	
IgG $n_{II}$ (+)						
IgM $n_{I}$ (+)	0; *p* = 0.855	0; *p* = 0.801	0; *p* = 0.886	0; *p* = 0.785	0; *p* = 0.838	
IgM $n_{II}$ (+)						

**Table 6 ijerph-18-04376-t006:** SARS-CoV-2 testing (RT-PCR) per 1000 individuals in two different regions of Poland.

	First Blood Collection ($n_{I}$)			Second Blood Collection ($n_{\mathrm{II}}$)		
	6 July	13 July	20 July	3 August	10 August	17 August
Silesia voiv.	37.4	40.3	43.3	49.7	55.6	60.2
Opole voiv.	13.8	15.1	16.3	19.0	20.8	22.9

## Data Availability

The data used to support the findings of this research are available from the corresponding author upon request.

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
