# Peer review of "SARS-CoV-2 Antibody Screening in Healthcare Workers in Non-Infectious Hospitals in Two Different Regions of Southern Poland (Upper Silesia and Opole Voivodeships): A Prospective Cohort Study"

_ijerph, 2021, doi:10.3390/ijerph18084376_

Round 1
Reviewer 1 Report
The authors RafaÅ‚ Jakub BuÅ‚dak et al in here evaluated the host serologic responses, measured with semi-quantitative ELISA tests (IgA, IgG, IgM abs) in sera of 90 individuals in Hospital no. 4 in Bytom, 84 HCWs in the University Hospital in Opole and 25 in a Miasteczko ÅšlÄ…skie local surgery. All volunteers had negative RT-PCR test results or not had the RT-PCR test performed within 30 days before sampling. What’s more, they found serological surveillance has the potential to provide more faithful information about virus contraction rate in HCWs for the first season of novel SARS-CoV-2 infections. The research focusing on COVID-19 is urgent and significant, and this is a topic of an incessant research interests. However, the manuscript also needs careful reversion to meet the high demands of IJERPH.
- Many reviewers have argued that this is a repeated research job. Compared to preciously published papers such as Barrett, E.S., Horton, D.B., Roy, J. et al. Prevalence of SARS-CoV-2 infection in previously undiagnosed health care workers in New Jersey, at the onset of the U.S. COVID-19 pandemic. BMC Infect Dis 20, 853 (2020). https://doi.org/10.1186/s12879-020-05587-2; Dimeglio, C., Herin, F., Miedougé, M., Cambus, J. P., Abravanel, F., Mansuy, J. M., Soulat, J. M., & Izopet, J. (2021). Screening for SARS-CoV-2 antibodies among healthcare workers in a university hospital in southern France. The Journal of infection, 82(1), e29–e32. https://doi.org/10.1016/j.jinf.2020.09.035; Folgueira, M. D., Muñoz-Ruipérez, C., Alonso-López, M. Á., & Delgado, R. (2020). SARS-CoV-2 infection in Health Care Workers in a large public hospital in Madrid, Spain, during March 2020. MedRxiv, 2020.04.07.20055723. https://doi.org/10.1101/2020.04.07.20055723; Garcia-Basteiro, A.L., Moncunill, G., Tortajada, M. et al. Seroprevalence of antibodies against SARS-CoV-2 among health care workers in a large Spanish reference hospital. Nat Commun 11, 3500 (2020). https://doi.org/10.1038/s41467-020-17318-x, the principle and innovation of this paper should be explained more in details.
- There are many abbreviations in use. Please show the full names before using them.
- The authors need to explain more and rewrite some details to avoid making readers confused.
- It’s necessary to add a clear hypothesis and aim for the study.
- Some fonts in the figure need to be adjusted.
Author Response
Thank you for the valuable advice and comment on the manuscript. In the following paragraphs we have given specific answers.
1. Many reviewers have argued that this is a repeated research job. Compared to preciously published papers such as:
- Barrett, E.S., Horton, D.B., Roy, J. et al. Prevalence of SARS-CoV-2 infection in previously undiagnosed health care workers in New Jersey, at the onset of the U.S. COVID-19 pandemic. BMC Infect Dis 20, 853 (2020). https://doi.org/10.1186/s12879-020-05587-2;
- Dimeglio, C., Herin, F., Miedougé, M., Cambus, J. P., Abravanel, F., Mansuy, J. M., Soulat, J. M., & Izopet, J. (2021). Screening for SARS-CoV-2 antibodies among healthcare workers in a university hospital in southern France. The Journal of infection, 82(1), e29–e32. https://doi.org/10.1016/j.jinf.2020.09.035;
- Folgueira, M. D., Muñoz-Ruipérez, C., Alonso-López, M. Á., & Delgado, R. (2020). SARS-CoV-2 infection in Health Care Workers in a large public hospital in Madrid, Spain, during March 2020. MedRxiv, 2020.04.07.20055723. https://doi.org/10.1101/2020.04.07.20055723;
- Garcia-Basteiro, A.L., Moncunill, G., Tortajada, M. et al. Seroprevalence of antibodies against SARS-CoV-2 among health care workers in a large Spanish reference hospital. Nat Commun 11, 3500 (2020). https://doi.org/10.1038/s41467-020-17318-x,
- the principle and innovation of this paper should be explained more in details.
We provide appropriate sentences mainly in the introduction, material and method as well as discussion sections (principles and innovation of this paper).
Introduction section:
“the present serological research was performed during the first wave of the rise of morbidity in Poland, where most institutions focused on genetic diagnosis of SARS-CoV-2 infection (vide RT-PCR) to eliminate infected health care workers (HCWs) from the health care system (HCS)”
“When estimating disease transmission in a specific occupational setting, only serological tests performed on two blood samples from the same volunteers 2 weeks apart can be fully reliable. Our serological assays were performed in so-called twin (even) sera to eliminate the risk of serological window, which significantly increases the diagnostic reliability of the present results. The majority of the published papers include serological data obtained from single serum sample collection (Garcia-Basteiro et al., 2020; Galan et al., 2020; Schneider et al., 2020; Trieu et al; 2020; Chen et al., 2020)”.
Material and Method section:
(Patients)
“We did not include volunteers who were RT-PCR positive in the analysis because 100% of (+) RT-PCR patients are IgG seropositive 10-19 days after infection (Guo et al., 2020, Long et al., 2020). Thus, we would have overestimated the numbers and we only wanted to screen active health care workers who did not know whether they had contracted SARS-CoV-2 infection (the social aspect of our study), especially since in addition to the scientific aspect of the result, the volunteers received a diagnostically reliable test result (double blood draw to avoid the risk of a serological window)”
(Antibody measurement)
“The ELISA test was made at two different time points (July/August 2020) with a 2-weeks gap between blood collections to avoid the “serological window” period”
Discussion section:
“Our data shows that focusing solely on the testing of individuals fitting a strict clinical case definition for COVID-19 will inevitably miss asymptomatic infected individuals”
“We observed the differences between the percentage distribution of IgG positive results in the University Hospital in Opole and Hospital no. 4 in Bytom (1.2% vs. 10% respectively). Employees of the hospital in Opole had been regularly tested with the RT-PCR test (every 4 weeks), unlike the employees of the other facilities surveyed, which could have had an impact on the identification of asymptomatic carriers of COVID-19”
“Our study included significantly more volunteers than qualified for the analysis. New infections, holiday periods or layoffs and absenteeism from work, two blood draws (with a 14-day break in between) resulted in a reduction in the final number of volunteers included in the analysis for reasons beyond the control of the researcher.”
We also provide grant numbers SUM, UO (inner funding).
2. There are many abbreviations in use. Please show the full names before using them.
R: The abbreviation paragraph has been updated and moved to the beginning of the manuscript to make their use more convenient.
3. The authors need to explain more and rewrite some details to avoid making readers confused.
R: The requiring wording has been corrected to be clearer. Language errors have been removed, the introduction has been updated with the latest data. The summary has also been redrafted to make it clearer for readers.
4. It’s necessary to add a clear hypothesis and aim for the study.
R: We provided clear hypothesis and aim for the study.
Here, we aimed to determine the serological status of HCWs who were not aware of prior SARS-CoV-2 infection or any other disease symptoms in selected non-infectious hospitals during the first wave of the epidemic in the southern region of Poland.
5. Some fonts in the figure need to be adjusted
R: The fonts in the figure have been adjusted to match the font of the main text.
Yours sincerely,
Rafal Buldak and co-authors
Reviewer 2 Report
I acknowledge that the authors have revised the work and the paper has improved since its first version.
The manuscript presents an important public health issue. Although the conclusion is relevant, it does not make any new contribution to existing knowledge so far. However, it reinforces the findings found in previous studies and the article is well constructed and conducted.
Author Response
Thank you for the valuable advice and corrections, which allowed the content of the final form of the manuscript to be improved. The article presents an important public health issue that is also a valuable component in antibody screening for healthcare professionals during the ongoing coronavirus pandemic. Linguistic errors in the text have been corrected, the introduction has been updated with the latest data.
Yours sincerely,
Rafal Buldak and co-authors
Reviewer 3 Report
The Authors did their best to address my remarks.
Author Response
Thank you for the valuable advice and corrections, it influenced the final shape and clarity of the manuscript. The article presents the subject of public health and aims to broaden the knowledge of antibodies in healthcare professionals. The wording of the conclusions has been improved, linguistic errors in the text have been corrected.
Yours sincerely,
Rafal Buldak and co-authors
This manuscript is a resubmission of an earlier submission. The following is a list of the peer review reports and author responses from that submission.
Round 1
Reviewer 1 Report
"The authors RafaÅ‚ Jakub BuÅ‚dak et al. in here present SARS-CoV-2 antibody screening in healthcare workers in a non-infectious hospital in two different regions of Southern Poland (Upper Silesia and Opole voivodeship). They found the IgG seropositivity of asymptomatic HCWs varied between 1.2% to 10% (Opole vs. Bytom, p<0.05). IgA seropositivity in HCWs was 8.8% in Opole and 7.14% in Bytom. IgM positive levels in HCWs in opole and Bytom was 1.11% vs. 2.38% respectively. What’s more, serological surveillance has the potential to provide more faithful information about virus contraction rate in HCWs for the first season of novel SARS-CoV-2 infections. The research related to SARS-CoV-2 is a topic of an incessant research interest which will attract many readers. In view of the relationships between SARS-CoV-2 antibody and data of healthcare workers in a non-infectious hospital are poorly understood, which represents a significant knowledge gap. Such knowledge is fundamental to the understanding of SARS-CoV-2. The objective of this project is to present the relationship between them. Addressing this critical knowledge gap will facilitate Covid-19 long-term goal of diagnosis and treatment. So, the manuscript is deserved to be published. However, it still needs much revision to meet the high demand for the journal. 1. Compared to preciously published papers such as Hansen, Johanna, et al. "Studies in humanized mice and convalescent humans yield a SARS-CoV-2 antibody cocktail." Science 369.6506 (2020): 1010-1014.; Lynch, Kara L., et al. "Magnitude and Kinetics of Anti–Severe Acute Respiratory Syndrome Coronavirus 2 Antibody Responses and Their Relationship to Disease Severity." Clinical Infectious Diseases 72.2 (2021): 301-308.; Chen, Yuxin, et al. "High SARS-CoV-2 antibody prevalence among healthcare workers exposed to COVID-19 patients." Journal of Infection 81.3 (2020): 420-426.; Chen, Yuxin, et al. "High SARS-CoV-2 antibody prevalence among healthcare workers exposed to COVID-19 patients." Journal of Infection 81.3 (2020): 420-426., the principle and innovation of this paper should be explained more in details. 2. There are many abbreviations in use. Please show the full names before use them. 3. There are some grammar errors which should be revised carefully. 4. It’s highly recommended to supply ethical information because ethical concern on a COVID-19 study is regarded to be emphasized. Especially, when we worried about, we have to insist on to be resolved."
Reviewer 2 Report
attached

Reviewer 3 Report
The manuscript presents an important public health issue. In conclusion, serological surveillance has the potential to provide more faithful information about virus contraction rate in HCWs.
Although this conclusion is relevant, it does not make any new contribution to existing knowledge so far.
In a quick bibliography search, we can immediately find numerous similar publications (see below for some extracted references). In the same sense, the Reviewer 2 has pronounced: "I suggest review manuscript because have repeat works".
This is the reason why I consider that this work does not make a significant contribution, beyond case studies in the same vein.
Otherwise, the article is well constructed and conducted.
Barrett, E.S., Horton, D.B., Roy, J. et al. Prevalence of SARS-CoV-2 infection in previously undiagnosed health care workers in New Jersey, at the onset of the U.S. COVID-19 pandemic. BMC Infect Dis 20, 853 (2020). https://doi.org/10.1186/s12879-020-05587-2
Dimeglio, C., Herin, F., Miedougé, M., Cambus, J. P., Abravanel, F., Mansuy, J. M., Soulat, J. M., & Izopet, J. (2021). Screening for SARS-CoV-2 antibodies among healthcare workers in a university hospital in southern France. The Journal of infection, 82(1), e29–e32. https://doi.org/10.1016/j.jinf.2020.09.035
Folgueira, M. D., Muñoz-Ruipérez, C., Alonso-López, M. Á., & Delgado, R. (2020). SARS-CoV-2 infection in Health Care Workers in a large public hospital in Madrid, Spain, during March 2020. MedRxiv, 2020.04.07.20055723. https://doi.org/10.1101/2020.04.07.20055723
Garcia-Basteiro, A.L., Moncunill, G., Tortajada, M. et al. Seroprevalence of antibodies against SARS-CoV-2 among health care workers in a large Spanish reference hospital. Nat Commun 11, 3500 (2020). https://doi.org/10.1038/s41467-020-17318-x
Isabel Galán, M., Velasco, M., Luisa Casas, M., José Goyanes, M., Rodríguez-Caravaca, G., Emilio Losa, J., Noguera, C., & Castilla, V. (2020). SARS-CoV-2 Seroprevalence Among All Workers in a Teaching Hospital in Spain: Unmasking The Risk. MedRxiv, 2020.05.29.20116731. https://doi.org/10.1101/2020.05.29.20116731
Schneider, S., Piening, B., Nouri-Pasovsky, P.A. et al. SARS-Coronavirus-2 cases in healthcare workers may not regularly originate from patient care: lessons from a university hospital on the underestimated risk of healthcare worker to healthcare worker transmission. Antimicrob Resist Infect Control 9, 192 (2020). https://doi.org/10.1186/s13756-020-00848-w
Trieu, M. C., Bansal, A., Madsen, A., Zhou, F., Sævik, M., Vahokoski, J., Brokstad, K. A., Krammer, F., Tøndel, C., Mohn, K., Blomberg, B., Langeland, N., & Cox, R. J. (2020). SARS-CoV-2-specific neutralizing antibody responses in Norwegian healthcare workers after the first wave of COVID-19 pandemic: a prospective cohort study. The Journal of infectious diseases, jiaa737. Advance online publication. https://doi.org/10.1093/infdis/jiaa737
Reviewer 4 Report
With interest, I read the manuscript ijerph-1087040.
Comments:
- The amount of work behind this report is huge.
- However, the aims are not clear. Even from the title, it is clear it is a study without a clear hypothesis.
- But even studies without hypothesis are fine, such as observational studies, if they make it possible to draw firm conclusions based on their results. In the case of this study, no clear conclusion can be made.
- What the Authors write in their “Conclusions” may be generally fine, i.e. that there is a need of careful HC personnel monitoring and protection, but it is not really supported by any data they created.
- The numbers are relatively small, especially considering seropositive individuals. Thus, any statistical calculations cannot yield solid and sound results.
- The number of statistical comparisons are enormous. They would probably be all non-significant if adjusted for multiple testing.
- What is “Collective analysis of both sera …” in Table 2-4? Why nI and nII do not sum up in Tables 2 and 4?
- What is the information from the analysis of PCR-negative subjects? PCR negativity means that they are not infectious anymore (even if they were before).
Other comments:
- What is “virus contraction rate”?
- Figure 3. “2nd” not “1st”, right?